

# BioDinamica: a toolkit for analyses of biodiversity and biogeography on the Dinamica-EGO modelling platform

Ubirajara Oliveira, Britaldo Soares-Filho, Rômulo Fernandes Machado Leitão and Hermann O. Rodrigues

Centro de Sensoriamento Remoto, Instituto de Geociências, Universidade Federal de Minas Gerais—UFMG, Belo Horizonte, Minas Gerais, Brazil

## ABSTRACT

Biogeography and macroecology are at the heart of the debate on ecology and evolution. We have developed the BioDinamica package, a suite of user-friendly graphical programs for analysing spatial patterns of biogeography and macroecology. BioDinamica includes analyses of beta-diversity, species richness, endemicity, phylodiversity, species distribution models, predictive models of biodiversity patterns, and several tools for spatial biodiversity analysis. BioDinamica consists of a sub-library of Dinamica-EGO operators developed by integrating EGO native functions with R scripts. The BioDinamica operators can be assembled to create complex analytical and simulation models through the EGO graphical programming interface. In addition, we make available "Wizard" tutorials for end users. BioDinamica can be downloaded free of charge from the Dinamica EGO submodel store. The tools made available in BioDinamica not only facilitate complex biodiversity analyses, they also help develop state-of-the-art spatial models for biogeography and macroecology studies.

## INTRODUCTION

Biogeographical and macroecological studies have multiplied largely over the last decade (*Ladle et al., 2015*). Proportionally, novel methods of analyses have also been developed. Many of these methods focus on spatial pattern representation, such as areas of endemism, species richness and beta-diversity (*Vilhena & Antonelli, 2015*; *Oliveira, Brescovit & Santos, 2015*), while others aim to predict these patterns (*Graham & Hijmans, 2006*; *Ferrier et al., 2007*). Similarly, there has been an increasing number of studies including phylogenetic trees due to the growing availability of data (*Hinchliff et al., 2015*) and hence the possibility of testing explicit evolutionary components through biogeography analyses. As a result, biogeographical and macroecological methods that apply phylogenetic data to understand evolutionary geographical patterns have also multiplied, e.g., phylogenetic beta diversity, phylogenetic endemism and phylodiversity (*Graham & Fine, 2008*; *Donnellan & Cook, 2009*). In addition to biogeographical and macroecological analyses, all these novel methods

Corresponding author
Ubirajara Oliveira,
ubirajara@csr.ufmg.br,
ubiologia@yahoo.com.br

are extremely important for conservation studies (*Whittaker et al., 2005*; *Mcgoogan et al., 2007*; *Fenker et al., 2014*).

There are few computer programs that make available in a single environment several analytical tools for biogeography and macroecology analyses—e.g., Passage (*Rosenberg & Anderson, 2011*). Even this software has only few available analyses, yet they do not directly involve maps. Software, like DIVA-GIS (*Hijmans et al., 2001*), that perform biogeographic analyses using maps remain rare. However, even DIVA-GIS package contains only a very limited set of functions. Most of the tools available for biogeography analyses are only present in R packages, which are difficult to be used by biologists or other specialists who do not master programming language. Friendly graphical interface programs are uncommon, and when available are limited to just a few specific analyses—e.g., for species distribution models (SDM) Open Modeller (*Souza Muñoz et al., 2011*), Maxent (*Phillips, Dudík & Schapire, 2004*), Modeco (*Guo & Liu, 2010*) and for biogeographical analyses PASSAGE (*Rosenberg & Anderson, 2011*) and DIVA-GIS (*Hijmans et al., 2001*).

Biogeography software to date do not encompass a wide set of relevant analyses. Some promising methods, such as Generalized Dissimilarity Model (GDM), for instance, (*Ferrier et al., 2007*) is eleven years old, but still little used (e.g., *Ferrier et al., 2012*; *Carnaval et al., 2014*; *Rosauer et al., 2014*), possibly because it is only available as a R package. Similarly, other methods, such as the Geographical Interpolation of Endemism—GIE (*Oliveira, Brescovit & Santos, 2015*), which identifies areas of endemism without the use of grid cells as sample units, have been barely used due to the absence of a friendly software—performing GIE requires a series of GIS standalone procedures. Even widely used methods, such as SDMs, have their functions dispersed in several R packages and various software. Moreover, processes required for modelling species distribution are often not available in SDM software, requiring the use of GIS and statistical software to perform a thorough analysis.

Given the growing interest in spatial analyses in biogeographic and macroecology, we have developed a set of user-friendly tools embedded in the Dinamica-EGO software (*Soares-Filho, Rodrigues & Follador, 2013*). Dinamica-EGO is a freeware (http://www.dinamicaego.com) for developing from simple to complex spatially explicit models, which has been applied to many environmental studies (see https://csr.ufmg.br/dinamica/publications). We coupled a series of Dinamica EGO operators with R code (*R Core Team, 2017*) to build more than 50 biogeographic and macroecological analytical functions (Table 1), all of which with a user-friendly graphical interface. These functions are stored in a sub library of Dinamica-EGO, named BioDinamica, thus allowing the user to build complex biodiversity models in a single integrated environment. In addition to the direct application of these tools to biogeography, biodiversity and macroecology (e.g., phylodiversity, species distribution models, phylogenetic endemism, areas of endemism, etc.), some of the available functions, such as generalized linear models (GLM), geographically weighted regression (GWR), and raster PCA projection (principal components analysis) are also applicable to several other study fields. BioDinamica takes advantage of Dinamica EGO high performance computing, nonetheless, requiring

computer resources as those available on common laptop computers, such as a minimum of 4GB of RAM and Windows or Linux operating system.

## SURVEY METHODOLOGY

### Overview

Functions provided include areas of endemism, species richness, phylodiversity, beta-diversity endemicity, species distribution models (SDMs), beta-diversity predictive models (GDM), interpolators, spatial analysis of ordination (PCA, PCR, NMDS), spatial statistical analysis (GLM, LM) and tools for conservation analysis, such as the Minimum Convex Hull (Table 1). All functions include R codes as well as specific R packages which are enveloped by the Dinamica EGO Operator called "Calculate R Expression". Although functions can be broken up for inspection, reuse, or further development, the users do not need to deal with the R code; instead they only need to configure or connect the parameters of these new hybrid operators by visually editing their inputs and outputs ports.

To facilitate the use of Biodinamica functions, we have standardized the operators' inputs (Fig. 1). Thus, functions for analyses of spatial diversity patterns (species richness, beta-diversity, areas of endemism, etc.) have as input a table in csv format with points of occurrence of species in three columns: sp, $x$ and $y$ (species name, longitude and latitude in decimal degrees) and a mask of the study area in shapefile format (Fig. 1). Analyses using phylogenetic data (phylodiversity, phylogenetic beta-diversity, phylogenetic-GDM, etc.) include a phylogenetic tree in *newick* format, along with the inputs for analyses of diversity pattern (species points and mask, as mentioned above). Analyses that rely on predictor variables (such as GDM, SDMs, interpolation and prediction by GLM, LM, SAR) use as input raster files only in the GeoTiff format. Spatial interpolation needs only a table in csv format with input variable values and respective geographic coordinates (columns: dependent, $x$ and $y$). Predictor-based interpolations (GLM, LM, GWR, SAR) use as input a table in csv format including the values of dependent variable and their coordinates (dependent, $x$, $y$), together with the raster predictor variables. All analyses outputs textual logs including specific statistics (Fig. 1). For analyses of spatial patterns, the functions output figures and graphs as well aimed to facilitate interpretation of results (Fig. 1). To use BioDinamica, one only needs to install Dinamica-EGO (http://csr.ufmg.br/dinamica/) and the package BioDinamica. Complete documentation is available at (http://csr.ufmg.br/dinamica/dokuwiki/doku.php?id=biodinamica) and online discussion list for questions and bugs at (https://groups.google.com/forum/#!forum/dinamica-ego). The online supplementary material of BioDinamica comes with BioDinamica installation guide and a guide that provides a brief explanation of its functions.

### Mapping spatial biodiversity patterns

BioDinamica includes several functions for spatial analyses of diversity patterns, such as beta-diversity, phylogenetic beta-diversity, endemicity, species richness, phylodiversity, and phylogenetic endemism. All these functions employ hexagonal tiles (equal area hexagons) as sample units, but also allow continuous interpolating of point data by using spatially explicit models. The interpolation models available in BioDinamica are the Spline method

Oliveira et al. (2019), *PeerJ*, DOI 10.7717/peerj.7213

Peer J

**Table 1** Description of main functions, inputs and outputs of BioDinamica.

| Tool group | Function name | Function | Inputs | Outputs | Reference |
|---|---|---|---|---|---|
| Biogeography | GIE—Geographic Interpolation of Endemism | Identify Areas of Endemism | Species points of occurence, map of study area | Raster maps, figures and reports | *Oliveira, Brescovit & Santos (2015)* |
| Biogeography | SCI—Species Composition Interpolation | Map beta-diversity and map partitions of beta-diversity (turnover and nestedness) | Species points of occurence, map of study area | Raster maps, figures and reports | *Oliveira, Vasconcelos & Santos (2017)* |
| Biogeography | PCI—Phylogenetic Composition Interpolation | Map phylogenetic beta-diversity and map partitions of beta-diversity (turnover and nestedness) | Species points of occurence, phylogenetic tree, map of study area | Raster maps, figures and reports | |
| Biogeography | SR—Species richness interpolation | Map species richness | Species points of occurence, map of study area | Raster maps, figures and reports | *Oliveira et al. (2019)* |
| Biogeography | RSR—Resampling of species richness interpolation | Resampling data for reduce effect of sampling differences to map species richness | Species points of occurence, map of study area | Raster maps, figures and reports | |
| Biogeography | GDM—Generalized Dissimilarity Model | Map beta-diversity by environmental predictors | Species points of occurence, map of study area and rasters of environmental predictors | Raster maps, figures and reports | *Ferrier et al. (2007)* |
| Biogeography | Phylo-GDM—Phylogenetic Generalized Dissimilarity Model | Map phylogenetic beta-diversity by environmental predictors | Species points of occurence, phylogenetic tree, map of study area and rasters of environmental predictors | Raster maps, figures and reports | *Rosauer et al. (2014)* |
| Biogeography | PD—Phylogenetic Diversity Interpolation | Map phylogenetic diversity | Species points of occurence, phylogenetic tree and map of study area | Raster maps, figures and reports | *Oliveira et al. (2019)* |

*(continued on next page)*

Oliveira et al. (2019), *PeerJ*, DOI 10.7717/peerj.7213

**Table 1** (*continued*)

| Tool group | Function name | Function | Inputs | Outputs | Reference |
|---|---|---|---|---|---|
| Biogeography | WE—Weight Endemism | Map of Weight endemism index by cell | Species points of occurence, map of study area | Raster maps, figures and reports | *Williams & Humphries (1994)* |
| Biogeography | PE—Phylogenetic Weight Endemism | Map of phylogenetic Weight endemism index by cell | Species points of occurence, phylogenetic tree and map of study area | Raster maps, figures and reports | *Rosauer et al. (2009)* |
| Biogeography | PS—Phylogenetic Spatialization | Map phylogentic information | Species points of occurence, phylogenetic tree and map of study area | Raster maps, figures and reports | |
| Biogeography | SRM—Species richness Model | Map species richness by model prediction (GLM, SAR, Universal Kriging) | Species points of occurence, map of study area and rasters of environmental predictors | Raster maps, figures and reports | |
| Biogeography | RSRM—Resampling of species richness Model | Resampling data for reduce effect of sampling differences to map species richness and predict values by model (GLM, SAR, Universal Kriging) | Species points of occurence, map of study area and rasters of environmental predictors | Raster maps, figures and reports | *Adapted from*: *Oliveira et al. (2019)* |
| Biogeography | PDM—Phylogenetic Diversity Model | Map phylogenetic diversity by model prediction (GLM, SAR, Universal Kriging) | Species points of occurence, map of study area and rasters of environmental predictors | Raster maps, figures and reports | |
| Biogeography | WEM—Weight Endemism Model | Map weight endemism index by model prediction (GLM, SAR, Universal Kriging) | Species points of occurence, map of study area and rasters of environmental predictors | Raster maps, figures and reports | *Adapted from*: *Williams & Humphries (1994)* |

Oliveira et al. (2019), *PeerJ*, DOI 10.7717/peerj.7213

**Table 1** (*continued*)

| Tool group | Function name | Function | Inputs | Outputs | Reference |
|---|---|---|---|---|---|
| Biogeography | PEM—Phylogenetic Weight Endemism model | Map phylogenetic weight endemism index by model prediction (GLM, SAR, Universal Kriging) | Species points of occurence, map of study area and rasters of environmental predictors | Raster maps, figures and reports | *Adapted from*: *Rosauer et al. (2009)* |
| Biogeography | Sampling Effort | Map density of samples | Sampling points and map of study area | Raster maps and figures | *Oliveira et al. (2019)* |
| Biogeography | SDM—Species Distribution Models | Modeling species distribution by environmental predictors and several algorithms of SDM | Species points of occurence, map of study area and rasters of environmental predictors | Raster maps, figures and reports | *Elith & Leathwick (2009)* |
| Biogeography | Niche overlap | Test niche overlap by SDM rasters | SDM rasters | Report table | *Broennimann et al. (2012)* |
| Biogeography | Minimum Convex Hull | Create minimum convex hulll polygon | Species points of occurence, map of study area | Raster maps | – |
| SDM tool | AUC—Area Under Curve | Calculate Area Under Curve statistic | Species points of occurence (presence and absence), raster with continuous values (SDM for instance) | Reports | *Hanley & McNeil (1982)* |
| SDM tool | Statistic for validation of Binary maps | Calculate precision, sensitivity, Kappa Cohen, Accuracy, Specificity and True Skill Statistic (TSS) | Species points of occurence (presence and absence), raster with binary values (SDM with threshold for instance) | Reports | *Stehman (1997)* |

Oliveira et al. (2019), *PeerJ*, DOI 10.7717/peerj.7213

**Table 1** (*continued*)

| Tool group | Function name | Function | Inputs | Outputs | Reference |
|---|---|---|---|---|---|
| SDM tool | Create Point samples | Create points based on binary maps. Convert SDM maps in poits of occurence for other analyses in BioDinamica | Species points of occurence (presence and absence), raster with binary values (SDM with threshold for instance) | Table | – |
| SDM tool | Sum of Maps | Calculate sum of maps in a folder | Folder with rasters (SDM binary, for example) | Rasters | – |
| SDM tool | Calculate area of Distributions | Calculate area of distributions based on binary maps of SDM | Raster with binary values (SDM with threshold for instance) | Table | – |
| SDM tool | Extract values to Points | Create a table with values of maps (rasters) based on spatial position of points | CSV with points $(x, y)$ | Table | – |
| SDM tool | Change Maximum and Minimum values | Rescale values of rasters to values between 0 and 1 | Raster | Rasters | – |
| Statistica and Ordination | Correlation | Calculate correlation between maps | Folder with rasters to analysis | Reports | – |
| Statistica and Ordination | Clustering of variables | Compute sets of variables with high correlation inside groups and low correlation betwwen them | Folder with rasters to analysis | Reports, tables and figures | *Chavent et al. (2012)* |
| Statistica and Ordination | Global Moran I | Compute Moran I index of spatial autocorrelation | Raster file | Reports | – |
| Statistica and Ordination | Spatial Variogram | Create graphic of variogram by distance | Raster file | Figure | – |

Oliveira et al. (2019), *PeerJ*, DOI 10.7717/peerj.7213

| Tool group | Function name | Function | Inputs | Outputs | Reference |
|---|---|---|---|---|---|
| Statistica and Ordination | Unsupervised Classification | Classification of rasters in clusters of pixels based on k-means, random forest and Clustering Large Applications | Folder with rasters to analysis | Rasters | *Kaufman & Rousseeuw (1990)* |
| Statistica and Ordination | PCA—Principal Component Analysis | Create rasters with axis of PCA based on variables (in raster format) | Folder with rasters to analysis | Rasters | *Hotelling (1933)* |
| Statistica and Ordination | PCA project— Principal Component Analysis for projection | Create rasters with axis of PCA based on variables (in raster format) and project to another scenario | Folder with rasters to analysis and folder with rasters for projection | Rasters | *Hotelling (1933)* |
| Statistica and Ordination | PCR—Principal Component Regression | Create rasters with axis of PCR analysis based on variables (in raster format) | CSV table with input points (samples) with value of dependent variable (continuous values) and folder with rasters to analysis | Rasters | *Jolliffe (1982)* |
| Statistica and Ordination | PCR project— Principal Component Regression for projection | Create rasters with axis of PCR analysis based on variables (in raster format) and project to another scenario | CSV table with input points (samples) with value of dependent variable (continuous values), folder with rasters to analysis and folder with rasters for projection | Rasters | *Jolliffe (1982)* |

Peer J

**Table 1** (*continued*)

| Tool group | Function name | Function | Inputs | Outputs | Reference |
|---|---|---|---|---|---|
| Statistica and Ordination | PLSR—Partial Least Squares Regression | Create rasters with axis of PLSR analysis based on variables (in raster format) | CSV table with input points (samples) with value of dependent variable (continuous values) and folder with rasters to analysis | Rasters | *De Jong (1993)* |
| Statistica and Ordination | PLSR project—Partial Least Squares Regression for projection | Create rasters with axis of PLSR analysis based on variables (in raster format) and project to another scenario | CSV table with input points (samples) with value of dependent variable (continuous values), folder with rasters to analysis and folder with rasters for projection | Rasters | *De Jong (1993)* |
| Statistica and Ordination | CPPLS—Canonical Powered Partial Least Squares | Create rasters with axis of CPPLS analysis based on variables (in raster format) | CSV table with input points (samples) with value of dependent variable (discrete values) and folder with rasters to analysis | Rasters | *Indahl, Liland & Naes (2009)* |
| Statistica and Ordination | CPPLS project—Canonical Powered Partial Least Squares | Create rasters with axis of CPPLS analysis based on variables (in raster format) and project to another scenario | CSV table with input points (samples) with value of dependent variable (discrete values), folder with rasters to analysis and folder with rasters for projection | Rasters | *Indahl, Liland & Naes (2009)* |

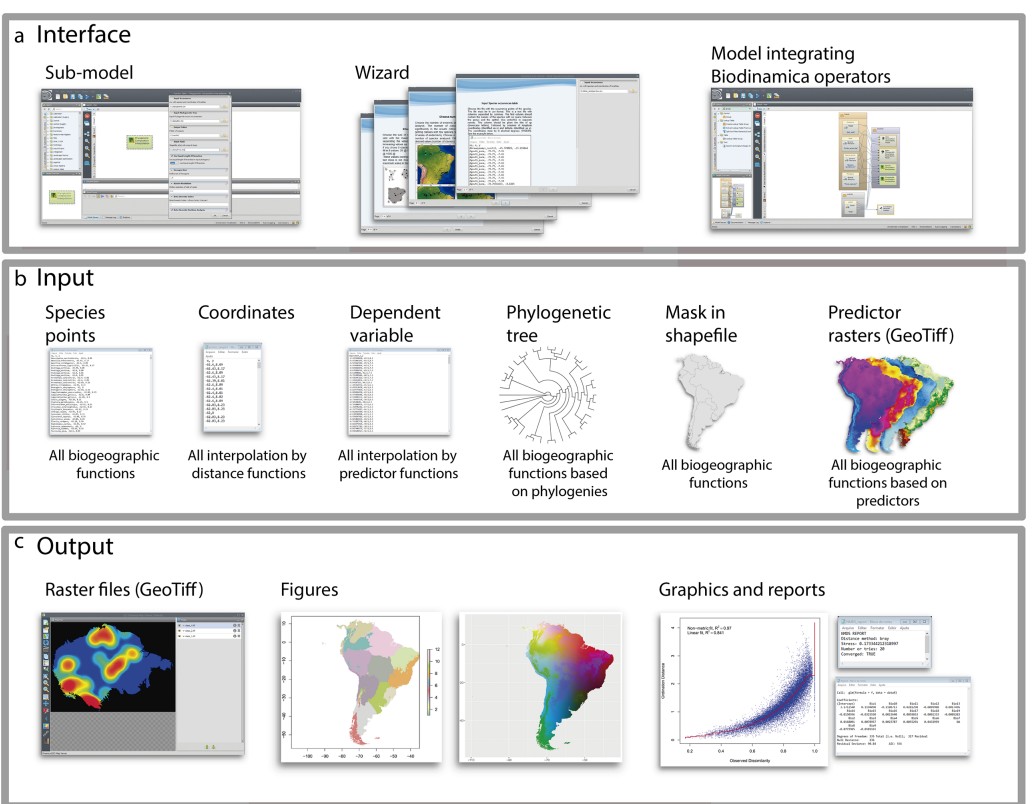

**Figure 1  Graphical interface.** (A) Interface; (B) inputs and (C) outputs of BioDinamica operators.

that derives a smooth prediction curve as a function of distance from observed points, nearest neighbour and the kriging, which applies a spatial interpolation according to a variogram distribution. Also available are analyses that predict spatial patterns (e.g., species richness, endemicity, phylodiversity) using predictor variables (e.g., climate variables) through generalized linear models (GLM), spatial autoregressive models (SAR), and universal kriging

Analyses of beta-diversity and phylogenetic beta-diversity patterns allow beta-diversity partitioning into two components, turnover and nestedness. These components can be represented by using either hexagonal tiles or continuous interpolation in order to visualize the spatial variation of each component. To map beta-diversity patterns, we have implemented GDM (*Ferrier et al., 2007*). This model predicts the beta-diversity patterns by using environmental predictors. Our implementation of GDM also allows applying the beta-diversity model to scenario modelling (past, or future, for example). Some diversity variables are more affected by sampling density and bias, such as species richness (*Oliveira et al., 2016*). To cope with that, we have implemented a rarefaction technique (*Oliveira et al., 2019*) that allows quantifying the relative richness between areas by standardizing sampling effort.

## Implementation of new methods

We also included novel analytical methods in BioDinamica. The Geographic Interpolation of Endemism (GIE) method identifies areas of endemism (AoE) (*Oliveira, Brescovit & Santos, 2015*). This method had not yet been fully implemented into a single integrated software environment. Hence, our GIE implementation needs not additional GIS software. The AoE outputs include raster maps for all scales and consensus, figures with AoE identification, tables describing how many and which species occur in each AoE, and a report with statistical information. This method can be used to identify patterns of congruent distribution among species for testing biogeographic hypotheses, such as vicariance, and for conservation priority studies (e.g., *Oliveira et al., 2019*).

To identify spatial patterns of beta-diversity, we have implemented a new method named Species Composition Interpolation (SCI) (*Oliveira, Vasconcelos & Santos, 2017*). This method spatially interpolates beta-diversity patterns by using values of a NMDS of the beta-diversity index matrix. Our implementation generates a raster map for each axis of the specialized NMDS, and a multiband raster cube for visualization of the axes through a RGB composite. The model also tests the spatial autocorrelation of the values of NMDS, which is a premise for this analysis. As another option, the user can classify the resulting maps into discrete regions (biogeographic regions) through techniques such as the k-means, random forest and CLARA (Cluster for large applications) unsupervised classification (*Ade & Hestir, 2017*). The latter technique allows choosing the number of classes, and then the algorithm identifies the intervals of values that best fit that number of classes (*Ade & Hestir, 2017*). We have also implemented an analysis analogous to SCI for phylogenetic beta-diversity and the Phylogenetic composition interpolation (PCI) (*Oliveira et al., 2019*).

## Evolutionary spatial patterns

BioDinamica provides a set of analytical tools for spatial mapping of evolutionary patterns. Using phylogenetic data, it is possible to create maps of phylogenetic beta-diversity (*Graham & Fine, 2008*), phylogenetic diversity, phylogenetic endemism (*Rosauer et al., 2009*), and to plot phylogenies on a map by using spatial interpolation. These analyses enable the user to map the evolutionary patterns of the groups studied in the geographic space, being useful for testing evolutionary hypotheses such as vicariance and dispersion across space. In addition, we have implemented the Phylogenetic generalized dissimilarity model (Phylo-GDM) (*Rosauer et al., 2014*). Finally, BioDinamica enables to perform scenario projections based on phylogenies analyses by using predictor interpolation (GLM, LM and SAR).

## Species distribution models

Today, species distribution models (SDM) are one of the most widely used biogeographic tools. We have implemented a set of SDM in BioDinamica (Bioclim, Boosted Regression Trees—BRT, Classification and regression trees—CART, Generalized Additive Models—GAM, Gradient boosting model—GBM, Generalized linear model—GLM, Mixture discriminant analysis—MDA, Multivariate adaptive regression splines—MARS, Recursive partitioning for classification trees—RPART, Maxlike—a maximum entropy tool—,
MAXENT, Random forest—RF and Support vector machines—SVM). In addition to SDMs, we have developed a set of ancillary tools for pre-processing and post-processing SDM inputs and outputs. These analyses allow modelling distribution of species by means of predictor variables (environmental variables). There is a wide range of uses for these models, from setting priorities for biodiversity conservation to testing of biogeographic and evolutionary hypotheses, such as events of niche divergence. In addition to SDMs themselves, we have created a set of tools for pre-processing and post-processing SDMs' inputs and outputs. For pre-processing, we have implemented two ways of creating pseudo-absences: the traditional one, which draws random points out of the presence samples of the species; and another based on sample evidence. The pseudo-absences based on sample evidence are obtained by sampling the pseudo-absences in the best-sampled regions (by using a kernel density map of sampling). In this way, the user provides occurrence points for the study group of species and the function generates a sampling effort map. From this map, the model draws samples (pseudoabsences) for areas more densely sampled. This technique is based on a simple premise: there is a greater probability that an absence is true when a well-sampled area (for a given taxonomic group) does not show occurrences of a particular species. In addition, various techniques for data validation and partitioning come with the SDMs package (Table 1).

## Interpolation

BioDinamica provides two forms of spatial interpolation: interpolation based on spatial data structure (spline, nearest neighbour and kriging); and statistical interpolation using predictive models (GLM, LM, SAR, GWR and universal kriging). Several biodiversity environmental data sets have an irregular spatial distribution and hence sampling gaps. To cope with that, spatial interpolation is used to produce continuous surfaces of these phenomena. For example, by using the Spline and Kriging spatial interpolation tools, we can interpolate continuous variables based only on their spatial autocorrelation structure as a predictor. For more complex problems, and where there is information on possible predictors, we can interpolate the spatial distribution by using generalized linear models (GLM). In addition to these methods, we have implemented hybrid models that employ the spatial structure of the predictive variables (Spatial autoregressive model: SAR). The BioDinamica analyses of biodiversity patterns (species richness, phylodiversity, endemicity, beta-diversity and phylogenetic beta-diversity) interpolate results by using either spline, nearest neighbour or kriging as an option. In addition, these patterns can also be interpolated by predictive models (GLM, SAR and universal kriging). The spatial and predictive interpolation techniques can be employed to test a wide range of biogeographic hypotheses, such as tests of patterns in biodiversity, as well as a means of filling in sampling gaps for biogeographic analyses.

## Statistical and ordination

Statistical analysis and ordering are central to biogeography and macroecology. To validate predictive models (such as SDM), we have built a binary map validation function. This function performs tests for accuracy, precision, sensitivity, specificity, Kappa and true skill
statistics (TSS). For continuous value maps, we have included the area under the curve (AUC). For the analysis of spatial patterns, we have included the analysis of Moran I and the Spatial Variogram.

One common problem in spatial modelling (including SDMs) is the high correlation between variables. To analyse the correlation between raster maps, BioDinamica includes a map correlation test and the Clustering of Variables analysis (*Chavent et al., 2012*). Another strategy to avoid correlation between predictor variables is by means of principal component analysis (PCA). In BioDinamica, we have implemented a function that creates a raster cube of the axes of PCA. These raster maps can be used as predictors because while they still represent the original variables, there is no more correlation between them. In order to use PCA raster in models designed for scenario projection (such as climate change scenarios), we have included the PCA projection option. This option employs the PCA model generated with the current variables to produce a PCA raster under a different scenario from the one whereby the variables were generated. Another implemented ordering technique produces raster maps of axes that are free of correlation assigning different weights to the variables to maximize their predictive ability. These ordination methods are principal component regression (PCR); partial least squares regression (PLSR); and canonical powered partial least square (CPPLS). In all of these techniques, the option of projection is available for scenario modelling. The raster maps generated from PCA, PCR and PLSR can be used as substitutes for the predictive variables in analyses in which the dependent variable has continuous values. The raster generated from PCA and CPPLS can be used as predictor variables in analyses in which the dependent variable has discrete values. Furthermore, we have implemented spatialization by non-metric multidimensional scaling NMDS. This function can be used to spatialize genetic data (genetic, phylogenetic or phylogeographic distance matrix) or even morphometric data (by the morphological distance matrix).

## General tools

BioDinamica provides a set of general tools for biogeography and macroecology analyses in an integrated modelling environment. In general, techniques employed in biogeographic analyses are only available as a series of standalone procedures in GIS software. For example, it is often necessary to cross-tabulate explanatory variables (such as climatic data) with species occurrence locations. BioDinamica "Extract values to points" function does this easily. In addition, the function "Create sample points" transforms binary maps (such as the ones from species distribution models) into occurrence points that can be input directly into other BioDinamica analytical tools.

## Proof of concept

For exemplifying the potential of BioDinamica, we use the software to explore three patterns of bird diversity in the Amazon: beta-diversity, species richness, and endemism by using as input the distribution polygons of bird species from Birdlife International (http://www.birdlife.org). *Oliveira, Vasconcelos & Santos (2017)* have already explored biogeographic patterns of Amazonian birds using museum data. Here, we test the

congruence of beta-diversity patterns as observed in *Oliveira, Vasconcelos & Santos (2017)* with those obtained from using occurrence polygons from the aforementioned dataset. We also investigate other bird geographical patterns (richness and index of endemism), which were not explored by *Oliveira, Vasconcelos & Santos (2017)*. In addition, we investigate the use of predictive models based on environmental variables (GLM and GDM) to spatially predict these biodiversity patterns.

The polygons of species distribution are converted into sample points through the function "Create samples" in BioDinamica by using 500 regular points per species. Points outside of the study area are ignored. We employ 179,188 records of 446 species of birds endemic to the Amazon. To spatially interpolate the sample data, we apply "Species composition interpolation" (SCI), "Species richness interpolation" (SR) and "Endemism by weighing endemism" (WE). All methods consist of spatial interpolation techniques. In addition, we apply "Generalized Dissimilarity model" (GDM) for beta diversity and "Generalized linear model" (GLM) for predicting species richness and endemism (SRM and WEM, respectively). For GDM analysis, we use hexagons as sample units (1 degree side) and the geographic distance from sample units for estimating the effects of the environmental covariates. In GLM, we use the Gaussian distribution for model estimation. In this analysis, we employ all the 19 climatic variables from Wordclim (http://www.worldclim.org/) as environmental predictors. For that, we convert these variables (related to temperature and rainfall) into axes of a principal component analysis (PCA) to remove the correlations between them and to reduce the number of variables. For this, we use only the first four axes of PCA, which account for ≈90% of the variance since they proved statistically significant, i.e., explaining more than expected by chance for the 19 variables (>5.26%).

Beta-diversity results show spatial patterns very similar to those observed for Amazonian birds through collection data (*Oliveira, Vasconcelos & Santos, 2017*). Interpolation (SCI) and prediction using environmental variables (GDM) are quite similar as well (Fig. 2). This is stressed by the high explanation of the model given the environmental variables (65% of explanation). This may indicate that the beta diversity geographic patterns as associated with the water basins of large rivers by *Oliveira, Vasconcelos & Santos (2017)* are, in fact, related to climatic conditions throughout the Amazon. Although all of these analyses are relatively complex, they are performed in a relatively short time. In a notebook with a 2.70 GHz Core i7 −7500U dual-core processor and 16GB of RAM, GDM runs in 3 min and 42 s and SCI in 27 min and 44 s.

The analysis of species richness outputs different results between using techniques of interpolation and prediction by GLM (Fig. 2). This can be caused by the low predictive capacity of the explanatory variables (climatic variables). The richness patterns resulting from interpolation (Fig. 2C) are very similar to those from analyses that only employ the distribution polygons of the species (see https://biodiversitymapping.org/wordpress/index.php/birds/). Thus, similarity between interpolated results with those observed in the polygon data, together with a large difference between interpolated results and the geographical distribution of environmental predictors, may indicate a low predictive power of the environmental variables for mapping bird richness patterns in the Amazon. The interpolated results more closely resemble the raw data from Birdlife International.

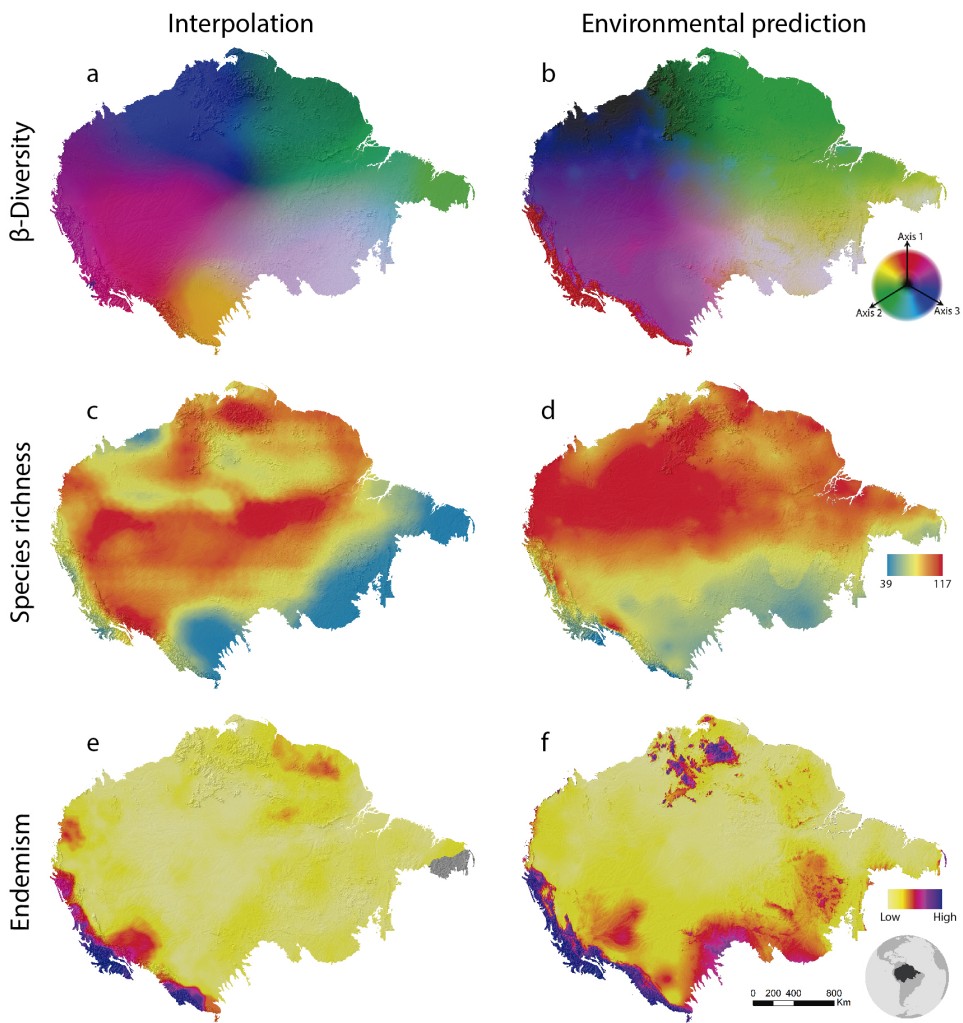

Interpolation                    Environmental prediction

**Figure 2  Amazonian bird diversity patterns based on Birdlife International data analysed through BioDinamica functions.** (A) Species composition interpolated by nearest neighbour, RGB represents the three axes of NMDS and (B) predicted by GDM; (C) species richness interpolated by nearest neighbour and (D) predicted by GLM; (E) weight endemism index corrected interpolated and (F) predicted by GLM.

However, this type of analysis requires validation using independent data to determine which patterns best reflect reality. The interpolation of species richness runs in 7 min and 50 s and the prediction by GLM in 4 min and 23 s.

The patterns of endemism (WE index) are consistent with that observed by *Oliveira, Vasconcelos & Santos (2017)* for areas of endemism. The areas identified with the highest number of species of most restricted distribution (Fig. 2) are coincident with the endemism areas identified by aforementioned authors who used a different set of data. However, we must be cautious with these similarities, since the patterns of endemicity (WE) are not necessarily congruent with the areas of endemism. Interpolation via WE runs in 15 min and 18 s and the prediction through GLM in 16 min.

The short computer time demonstrates the efficiency of BioDinamica in processing large datasets. In addition, BioDinamica allows compressing the dimensionality of predictor variables through the "PCA function". Many other biogeographic patterns analyses are also doable using this same dataset and other BioDinamica functions, such as GIE, phylogenetic endemism, phylogenetic beta diversity, etc. This in turn demonstrates the software versatility in exploring geographical patterns of biological data.

### Wizard: a tutorial interface

All the functions of BioDinamica are available as graphical operators of Dinamica-EGO. In addition, we provide model examples containing wizard tutorial. In this way, the user is guided through an illustrated tutorial that helps setting up and running the BioDinamica functions. Not only wizard tutorial illustrates applications, it also facilitates access to literature references (Fig. 1). Lastly, BioDinamica installation comes with sample datasets for training. Also available is an online guidebook with a comprehensive tutorial on all BioDinamica functions (http://csr.ufmg.br/dinamica/dokuwiki/doku.php?id= biodinamica).

## CONCLUSIONS

BioDinamica encompasses a wide variety of tools for spatial analyses of biodiversity, biogeography, macroecology and evolution. Developed using Dinamica-EGO freeware, BioDinamica delivers high performance on a user-friendly interface. In particular, the Dinamica-EGO platform allows the use of all functions into more complex models that includes loops, iterations and bifurcation pipelines. In this way, the BioDinamica functions become components of advanced models for conservation analyses and environmental simulations developed by using EGO graphical programming language.

## ACKNOWLEDGEMENTS

We thank Adalberto J. Santos, Ignacio Avila, Leonardo Sousa Carvalho, Marcelo Leandro Bueno, William Leles de Souza Costa for ideas, motivation and testing BioDinamica.

### Funding

Ubirajara Oliveira and Britaldo Soares Filho received support from CNPQ, FAPEMIG, Climate and Land Use Alliance, the World Bank, and the Humboldt Foundation. The funders had no role in study design, data collection and analysis, decision to publish, or preparation of the manuscript.

### Grant Disclosures

The following grant information was disclosed by the authors:
CNPQ.
FAPEMIG.

Climate and Land Use Alliance.

World Bank.

Humboldt Foundation.

## Competing Interests

The authors declare there are no competing interests.

## Author Contributions

- Ubirajara Oliveira conceived and designed the experiments, performed the experiments, analyzed the data, contributed reagents/materials/analysis tools, prepared figures and/or tables, authored or reviewed drafts of the paper, approved the final draft.
- Britaldo Soares-Filho, Rômulo Fernandes Machado Leitão and Hermann O. Rodrigues conceived and designed the experiments, performed the experiments, analyzed the data, contributed reagents/materials/analysis tools, authored or reviewed drafts of the paper, approved the final draft.

## Data Availability

BioDynamica is available at http://csr.ufmg.br/dinamica/dokuwiki/doku.php?id= biodinamica.

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
