# Peer review of "BioDinamica: a toolkit for analyses of biodiversity and biogeography on the Dinamica-EGO modelling platform"

_PeerJ, doi:10.7717/peerj.7213_

## Round 0.1 · original submission · Major Revisions

I find that you manuscript may be accepted, if you accept the suggestions made by the reviewers. I hope to see a revised version as soon as possible.

Best regards

Juan Morrone

·

Basic reporting

The authors have developed a new software platform called BioDinamica, which encompasses more than 50 user-friendly functions related to biogeography and macroecology. I commend the authors for the integration of multiple biogeographical and macroecological analyses into one package, with a user-friendly interface. One aspect that deserves to be mentioned is the inclusion of recent novel techniques into the platform, such as geographic interpolation of endemism or species composition interpolation (although the latter simply consists in plotting ordination axes values in a map using an RGB composite, and thus relatively easy to code in R). Also, sufficient background in the references is provided, although more information on the platform in which BioDinamica is embedded (Dinamica-EGO) is needed. This is relevant in this context, as it seems not to be a widespread or widely known software. The manuscript is overall clearly written in professional, unambiguous language.

Some comments on tables and figures are as follows: Table 1 is fine, but please provide references for each method in a new column. The high number of functions make it virtually impossible to describe each of them in the text, so it is important to provide the user relevant information about each analysis. Figure 1 is poorly relevant in the context of the software, given it is a merely description of the interface, inputs and outputs. I suggest redoing this figure using a flowchart instead. There are many examples of this kind of figures in the literature related to software development. A numerical scale in Figure 2 should be included, at least for species richness.

Some minor spelling mistakes are as follows: The citation is 2018 or in press at line 103? It should be “Phylogenetic GDM” at line 108. At Table 1, correct the spelling in the SDM tool “Calculate area of Disdributions” and in the Statistica and Ordination tool “Gloabl Moran I” and “Unsupervisioned Classification by K-means”.

Experimental design

The research aim is well defined, and it is also stated how the research fills a gap in biogeography and macroecology. However, I am not fully convinced about including SDM tools, as there are several platforms that enable integrative analyses of species distributions. See, for instance, Guo & Liu (Ecography, 2010, V33, pp 637-642), Naimi & Araújo (Ecography, 2016, V39, pp 368-375) and Cobos et al. (PeerJ, 2019, V7, e6281). I think the authors should make an extra effort to stand out from the rest of software already available. The package could include some novel algorithms for model fitting, like random forests, support vector machines, neural networks, among others. See Naimi & Araújo (Ecography, 2016, V39, pp 368-375).

The methods described have sufficient information, but more detail is needed at some times, particularly for those methods recently developed. Could you describe this rarefaction technique in more detail at lines 77-78? It is not clear whether this is a new method or refers to classical rarefaction, because you have cited an unpublished study. I also suggest that you let the user to choose the ordination technique at lines 97-98, instead of constraining them to use only NMDS. Eigenanalyses are valuables per se, as they give the proportion of variance explained by the axes. In addition, using three axes sounds restrictive. Users should base their decision of the number of axes chosen on the stress.

I also suggest that you improve the description about k-means at lines 101-102. Although k-means is a supervised learning technique, the user can choose the optimal number of clusters a posteriori (using, for instance, the “elbow method”). Is this kind of implementation present or is the user restricted to a priori define the number of clusters? Please provide more detail about the SDMs implemented at lines 112-113. Moreover, indicate the algorithm used for spatial thinning at lines 117-119. The benefit of the technique presented at lines 119-121 is not clear. Indeed, how does this spatial bias could properly reflect the available environmental conditions? Is there evidence of a better performance relative to extracting random background points?

Lobo et al. (Glob. Ecol. Biogeogr. V17, pp 145-151) have shown that the AUC method you have described at lines 140-142 has several drawbacks that prevent its use in SDMs and may be misleading. Please warn about this, and consider including other model performance metrics, such as the AIC. See, for example, Warren & Seifert (Ecol. Applic., 2011, V25, pp 331-342) and Radosavljevic et al. (2014, J. Biogeogr. V41, pp 629-643). Finally, please justify why you considered only 3 dimensions at lines 147-148, as one or two could eventually suffice. Can the user select one or two components?

All the data, information and tutorials for a reproducible research are available at the link provided in the text.

Validity of the findings

Conclusions are, in general, well stated, linked to original research question and limited to supporting results. Sometimes, nevertheless, the authors should explain in more detail certain results. For instance, an explanation for the strong difference between both patterns are not given at lines 195-199. Besides, there is congruence between the pattern depicted in Fig. 2e and that of Fig. 2 in Oliveira et al. (Sci. Rep., 2017, V7, 2992). Fig. 2f, by contrast, also shows high endemism areas in the north and the south east. Authors should explain this difference at lines 200-204.

Additional comments

Overall, the manuscript, software and data are sound, and provide a user-friendly interface software with numerous biogeographical and macroecological analyses. After the authors have taken my comments into consideration, I believe the study merits publication. Thank you for letting me reviewing this manuscript and have a good review.

·

Basic reporting

No comments.

Experimental design

No comments.

Validity of the findings

No comments.

Additional comments

The paper presents an interesting toolkit for dealing with biodiversity and biogeography analyses based on an a user-friendly graphical interface. BioDinamica will certainly be a useful tool for the whole community of biogeographers, conservacionists, taxonomists and ecologists.
The text is straight-forward, and its goals are well defined.
In this sense, I recommend accepting the paper without modifications.

·

Basic reporting

This paper introduces BioDinamica, a free package with a user-friendly interface that reunites algorithms for several spatial analyses for biogeography and macroecology. Such a package is very welcome, since many of these algorithms are already available but dispersed in different software, some of which lack a GUI or are paid. The package seems to be very complete and flexible. Each analysis has an associated wizard which guides the user through the analysis, which greatly facilitates the use of the program.

The manuscript itself is concise, well-written and clearly structured. I have pointed some suggestions for improvement directly in the text file. There are some main points that should be reviewed:

1) Introduction: before you introduce your own software, it would be enlightening to present a short review on already available packages and software. Surely most of the analysis included in BioDinamica can also be performed elsewhere (an exception is GIE). You should explain if those packages/software have any shortcomings, and how BioDinamica is a solution. Are they paid software? Are they all command-line-based? Are the needed functions dispersed in many different platforms and software?

2) You should include a section mentioning the utilities, such as "Extract values to points", which are very simple using BioDinamica compared with GIS software, for example. I also have a suggestion for future versions: including a simple "Join" function to extract names from polygon shapefiles to points (to recover names from administrative units, conservation units, etc). This would be useful to check the accuracy of the points.

3) I feel that the text lacks a biological “spin” when introducing the different analysis. For instance, your SDM section reads “Species Distribution models: Today, species distribution models (SDM) are one of the most widely used biogeographic tools.” This does not make clear why a biologist might want to estimate a SDM. The paper would be greatly enhanced if each section started with a short statement on what it is good for. There is no need to give too much detail nor to review literature, but just say what a biologist might gain from estimating SDMs, interpolating betadiversity, etc. This is especially true for the guidebook: it lacks biology altogether. I strongly advise that the next version of the manual includes a short statement for each analysis explaining what it is useful for.

4) Some of the methods used in the proof of concept are unclear and need further clarification, such as the generation of records from polygons, number of PCA axes used, etc. I have explained my doubts directly in the text.

5) One of the greatest strengths of similar software I am used to (e.g. MrBayes, Mesquite, BEAST) is the existence of an online group or email list which joins the community of users with the developers. As BioDinamica gets more popular, many end-users will undoubtedly have difficulties with running some of the more complex analysis, or even find bugs, and it would be great if they have something other than the manual to turn to. I advise you to set up and online forum or email list (BEAST use Google Groups, MrBayes uses SourceForge, Mesquite has a mail list) and already mention it in the paper.

6) Figure 2: this figure is already very informative but I think it can be substantially improved. First, it lacks a “column” with raw data: the non-interpolated values of e.g. species richness estimated from the points. Without seeing the raw data, I find it very difficult to evaluate if your interpolation makes sense or not. It was also annoying to compare it with Oliveira et al. 2017 because I had to go back and forth between the two papers. I suggest that you remake the figure with 6 columns (or maybe 2 figures with 3 columns): 1- raw species richness estimated from samples of polygons of Birdlife, 2- its interpolated values and 3- and its environmental prediction, 4- raw species richness estimated from museum specimens (not present in Oliveira et al. 2017!), 5- its interpolated values and 6- its environmental prediction.

Apart from the manuscript, I also tried downloading BioDinamica and running some of the analysis. I have some comments:

7) You should include links to the pages of Dinamica Ego 4.0.11 and Enhancement Plugin directly in the BioDinamica website, since you have to download both anyway.

8) At first, my input data had some errors in the coordinates, which made BioDinamica indicate an error. If possible, the next version should indicate the line containing the record where the error was found, otherwise it may be difficult to find the mistake.

9) GIE: I could run successful analyses with 5–7 classes (in a very timely manner – only a few minutes for a global dataset), but the analysis with 10 classes failed (“atributo ‘names[2] deve ter o mesmo comprimento que o vetor [1]” – by the way, it would be best if these error reports are displayed in English). I also had problems running an analysis into the same folder with output results. You should include an option for overwriting results or checking if the output folder already has data. This might be true for other analysis as well.

10) Minimum convex hulls: I got the message "Script X depends on submodels that could not be found in your system. Would you like to install and download from the Online Repository?". After I agreed to install the submodels, I could run the analysis, but the tiff files were empty – I’m not sure what went wrong.

11) Extract values per points: I got the error message: “wc2.0_bio_10m_01 is not a valid column name” but all values seem to have been correctly extracted. This analysis is much more straightforward and quick than using GIS software (the other way I would have done it).

12) SDM: after I input the occurrences I got the error "the functor "Sdm" functor cannot be executed. Its purpose is to replace a missing or incompatible submodel functor previously used in this script". I could not run this analysis.

13) Species richness interpolation: first I tried to interpolate using a world map as a mask and got the error “Error caught in R execution: ‘Topology Exception: input geom 0 is invalid: self-intersection at or near point xxx yyy at zzz www.” Then I tried to do the same analysis using an Argentina map as mask; this analysis advanced a little more, but then I got the error “the real value –nan(ind) is not valid”. I could not run this analysis.

I did not spend too much time trying to troubleshoot the errors I mentioned above – it could be something with my data or with the parameters I set. Anyway, this clearly demonstrates why it would be good to have an online group for this kind of situation.

The guidebook is welcome but needs several improvements. As mentioned above, each section should start with a quick practical/biological explanation of the purpose of the analysis. It also lacks appropriate references. For instance, each SDM should have its reference very explicitly included (e.g. the articles describing BioClim, SVM, MaxEnt, etc). This is also true for the Wizards: they MUST start with something like “Users using this analysis should cite papers X and Y” – for instance, the SDM Wizard currently cites a general review on SDMs). The guide could also include examples of successful outputs. Strangely, I could not use ctrl + F to search for words and navigate the PDF – maybe some problem with converting the file to PDF? Finally, the guide is in dire need of revision of the English – there are still many words and expressions in Portuguese and some portions are poorly translated. I strongly advise including all references in Wizards and preparing a revised version of the guidebook before the paper is published.

In sum, this is a nice and concise paper that presents a package which has the full potential to become one of the main tools in biogeography and macroecology. The software itself is easy to use, and although I encountered some problems while running some analyses, I am confident this can be solved with an online group for troubleshooting or addressed in future versions of the software. The functions that I could run were very easy and quick compared to other software that perform the same task. This seems like the start of a large project that has a great potential to be improved and expanded over the years.

Best regards,
Ivan L. F. Magalhaes

Experimental design

No comment

Validity of the findings

No comments

---

## Round 0.2 · accepted · Accept

Revisions have been made, so your paper is now accepted.

Best regards

Juan

#